# MEAN-FIELD ANALYSIS OF BATCH NORMALIZATION

## ABSTRACT

Batch Normalization (BatchNorm) is an extremely useful component of modern neural network architectures, enabling optimization using higher learning rates and achieving faster convergence. In this paper, we use mean-field theory to analytically quantify the impact of BatchNorm on the geometry of the loss landscape for multi-layer networks consisting of fully-connected and convolutional layers. We show that it has a flattening effect on the loss landscape, as quantified by the maximum eigenvalue of the Fisher Information Matrix. These findings are then used to justify the use of larger learning rates for networks that use BatchNorm, and we provide quantitative characterization of the maximal allowable learning rate to ensure convergence. Experiments support our theoretically predicted maximum learning rate, and furthermore suggest that networks with smaller values of the BatchNorm parameter $\gamma$ achieve lower loss after the same number of epochs of training.

## 1 INTRODUCTION

Deep neural networks have achieved remarkable success in the past decade on tasks that were out of reach prior to the era of deep learning (Krizhevsky et al., 2012; He et al., 2016b). Amongst the myriad reasons for these successes are powerful computational resources, large datasets, new optimization algorithms, and modern architecture designs (Russakovsky et al., 2015; Kingma & Ba, 2015). In many modern deep learning architectures, one key component is batch normalization (BatchNorm). BatchNorm is a module that can be introduced in layers of deep neural networks that normalizes hidden layer outputs to have a common first and second moment. Empirically, BatchNorm enables optimization using much larger learning rates, and achieves better convergence (Ioffe & Szegedy, 2015).

Despite significant practical success, a theoretical understanding of BatchNorm is still lacking. A widely held view is that BatchNorm improves training by "reducing of internal covariate shift" (ICF) (Ioffe & Szegedy, 2015). Internal covariate shift refers to the change in the input distribution of internal layers of the deep network due to changes of the weights. Recent results (Santurkar et al., 2018), however, cast doubt on the ICF expalanation, by demonstrating that noisy BatchNorm increases ICF yet still improves training as in regular BatchNorm. This raises the question of whether the utility of BatchNorm is indeed related to the reduction of ICF. Instead, it is argued by Santurkar et al. (2018) that BatchNorm actually improves the Lipschitzness of the loss and gradient.

Meanwhile, dynamical mean-field theory (Sompolinsky & Zippelius, 1982), a powerful theoretical technique, has recently been applied by Poole et al. (2016) to ensembles of multi-layer random neural networks. This theory studies networks with an i.i.d. Gaussian distribution of weights and biases. Most recent work focuses on the analysis of order parameter flows and their fixed points (Schoenholz et al., 2017; Xiao et al., 2018; Yang & Schoenholz, 2017), including their stability and decay rates. Importantly, Karakida et al. (2018) also successfully used mean-field analysis to estimate the spectral properties of the Fisher Information Matrix.

In this paper, we analytically quantify the impact of BatchNorm on the landscape of the loss function, by using mean-field theory to estimate the spectral properties of the Fisher Information Matrix (FIM) for typical batch-normalized neural networks. In particular, it is shown that BatchNorm reduces the maximal eigenvalue of the FIM provided that the normalization coefficient $\gamma$ is not too large. By drawing on results linking Fisher Information to the geometry of the loss function, we explain how BatchNorm neural networks can be trained with a larger learning rate without leading to parameter

explosion, and provide upper bounds on the learning rate in terms of the BatchNorm parameters. As an additional contribution motivated by our theoretical findings, we demonstrate an empirical correlation between the BatchNorm parameter $\gamma$ and test loss. In particular, networks with smaller $\gamma$ achieve lower loss after a fixed number of training epochs.

## 2 PRELIMINARIES

In our theoretical analysis, we employ the recent application of mean-field theory to neural networks which studies an ensemble of random neural networks with pre-defined i.i.d. Gaussian weights and biases. In this section, we provide background information and briefly recall the formalism of Karakida et al. (2018) which first computes spectral properties of the Fisher Information of a neural network and then relates it to the maximal stable learning rate.

### 2.1 FISHER INFORMATION MATRIX AND LEARNING DYNAMICS

Given a data distribution $\mathcal{D}$ over the set of instance-label pairs $\mathcal{X} \times \mathcal{Y}$, a family of parametrized functions $f_\theta : \mathcal{X} \to \mathcal{Y}$ and a loss function $l(f, y)$, our focus will be to ensure convergence of the following gradient descent with momentum update rule:

$$\theta_{t+1} = \theta_t - \eta \nabla L(\theta_t) + \mu(\theta_t - \theta_{t-1}) \ , \tag{1}$$

where $L(\theta)$ is the unobserved population loss,

$$L(\theta) := \mathbb{E}_{(x,y)\sim\mathcal{D}} \big[ l(f_\theta(x), y) \big] \ . \tag{2}$$

In practice, the parameters are determined by minimizing an empirical estimate of equation 2 using a stochastic generalization (SGD) of the update rule equation 1. We neglect this difference by always working in the asymptotic limit of large sample size and moreover assuming full-batch gradient updates.

Suppose the loss function can be expressed in terms of a parametric family of positive densities as $l(f_\theta(x), y) =: -\log p_\theta(x, y)$. This assumption holds true for a large class of losses including squared loss and cross-entropy loss. Let $I_\theta$ denote the Fisher Information Matrix (FIM) associated with the parametric family induced by the loss,

$$I_\theta := \mathbb{E}_{(x,y)\sim\mathbb{P}_\theta} \left[ \nabla_\theta \log p_\theta(x, y) \otimes \nabla_\theta \log p_\theta(x, y) \right] \ , \tag{3}$$

where $\otimes$ denotes Kronecker product and $\mathbb{P}_\theta$ denotes the probability distribution over $\mathcal{X} \times \mathcal{Y}$ with density $p_\theta(x, y)$. Recall that under suitable regularity conditions the following identity holds:

$$I_\theta = -\mathbb{E}_{(x,y)\sim\mathbb{P}_\theta} \left[ \operatorname*{Hess}_\theta \log p_\theta(x, y) \right] \ , \tag{4}$$

where $\operatorname{Hess}_\theta$ denotes the Hessian with respect to $\theta$. The above right-hand side is closely related to the Hessian of the population loss,

$$\operatorname{Hess}\big(L(\theta)\big) = -\mathbb{E}_{(x,y)\sim\mathcal{D}} \left[ \operatorname*{Hess}_\theta \log p_\theta(x, y) \right] \ , \tag{5}$$

where we interchanged the Hessian with the expectation value. In fact, if we assume that the estimation problem is well-specified so that there exist parameters $\theta_*$ such that the data distribution is generated by $\mathbb{P}_{\theta_*} = \mathcal{D}$, then we obtain the following equality between the Hessian of the population loss and the FIM evaluated at the optimal parameters,

$$\operatorname{Hess}\big(L(\theta_*)\big) = I_{\theta_*} \ . \tag{6}$$

If $\theta$ is initialized in a sufficiently small neighborhood of $\theta_*$, then by expanding the population loss $L(\theta)$ to quadratic order about $\theta_*$ one can show that a necessary condition for convergence is that the step size is bounded from above by (LeCun et al., 2012; Karakida et al., 2018)[1],

$$\eta < \eta_* := \frac{2(1+\mu)}{\lambda_{\max}\big(\operatorname{Hess}(L(\theta_*))\big)} = \frac{2(1+\mu)}{\lambda_{\max}(I_{\theta_*})} \ , \tag{7}$$

---

[1]In the quadratic approximation to the loss, the optimal learning rate is in fact $\eta_*/2$.

where $\lambda_{\max}(M)$ denotes the largest eigenvalue of the matrix $M$. Rather than computing the optimal parameters $\theta_*$ directly, we follow the strategy of Karakida et al. (2018) by estimating the following quantity and arguing that the distribution of the weights and biases is not significantly impacted by the training dynamics,

$$\bar{\lambda}_{\max} := \mathbb{E}_{\theta}\left[\lambda_{\max}\left(I_{\theta}\right)\right] \quad, \tag{8}$$

where $\mathbb{E}_{\theta}$ denotes the expectation value with respect to the weights and biases. This heuristic was shown to yield a remarkably accurate prediction of the maximal learning rate in (Karakida et al., 2018).

In this paper we adopt the data modeling assumption that the joint density factors as $p_{\theta}(x, y) = p(x)p_{\theta}(y \mid x)$ where $p(\cdot)$ denotes the probability density of the marginal distribution of the covariates, which is independent of $\theta$. Under this factorization assumption, the FIM simplifies to

$$I_{\theta} = \mathbb{E}_{(x,y)\sim\mathbb{P}_{\theta}}\left[\nabla\log p_{\theta}(y \mid x) \otimes \nabla_{\theta}\log p_{\theta}(y \mid x)\right] \quad. \tag{9}$$

Focusing on the Gaussian conditional model $p_{\theta}(y|x) \propto \exp(\frac{1}{2}\|f_{\theta}(x) - y\|_2^2)$, the FIM further simplifies to

$$I_{\theta} = \mathbb{E}_{x\sim\mathcal{D}}\left[\nabla_{\theta}f_{\theta}(x) \otimes \nabla_{\theta}f_{\theta}(x)\right] \quad. \tag{10}$$

The family of parametrized functions $f_{\theta} : \mathbb{R}^{N_0} \to \mathbb{R}^{N_L}$ is chosen to be the family of functions computed by a multi-layer neural network architecture with $N_0$ input nodes, $N_L$ output nodes and $L \geq 1$ layers. In this paper, we consider neural networks consisting of fully-connected (FC) and convolutional (Conv) layers, with and without batch normalization. The pointwise activation is denoted by $\sigma$, which is taken to be the rectified linear unit (ReLU) in this paper. Our analysis can be straightforwardly extended to other architectures and non-linearities. We use $h_{\theta}^l(x)$ to denote the output of layer $l$ and the input to layer $l + 1$. Clearly we have $h_{\theta}^0(x) = x$ and $h_{\theta}^L(x) = f_{\theta}(x)$.

## 3 THEORY

In this section we focus on applying dynamical mean-field theory to study the effect of introducing batch normalization modules into a deep neural network by estimating the largest eigenvalue of the FIM. This estimate, in turn, provides an upper bound on the largest learning rate for which the learning dynamics is stable. This section is structured as follows: We first define various thermodynamic quantities (order parameters, 6 for fully-connected layers and 9 for convolutional layers) that satisfy recursion relations in the mean-field approximation. Then we present an estimate of $\bar{\lambda}_{\max}$ in terms of these order parameters, generalizing a result of Karakida et al. (2018). Using this estimate, we study how $\bar{\lambda}_{\max}$ and $\eta_*$ are affected by BatchNorm and calculate their dependence on the Batch-Norm coefficient $\gamma$. Detailed derivations of the order parameters, their recursions, and the associated eigenvalue bound are deferred to the Supplementary Material.

### 3.1 FULLY CONNECTED LAYERS

A general fully connected layer with input activation $h^l(x)$ and output pre-activation $z^{l+1}(x)$ is described by the affine transformation,

$$z^{l+1}(x) := W^{l+1}h^l(x) + b^{l+1} \quad, \tag{11}$$

where $W^{l+1} \in \mathbb{R}^{N_{l+1}\times N_l}$, $b^{l+1} \in \mathbb{R}^{N_{l+1}}$ and $N_l$ denotes the number of units in layer $l$. In the framework of mean-field theory, we will consider an ensemble of neural networks with Gaussian random weights and biases distributed as follows,

$$[W^{l+1}]_{ij} \sim N(0, \sigma_{\mathrm{w}}^2/N_l) \quad, \qquad b_i^{l+1} \sim N(0, \sigma_{\mathrm{b}}^2) \quad. \tag{12}$$

In the case of a standard fully connected layer, the input activation satisfies the recursions $h^l(x) = \sigma(z^l(x))$, where $\sigma$ denotes the pointwise activation.

A batch-normalized fully connected layer, in contrast, satisfies the following recursion,

$$h^l(x) := \sigma\left(\frac{z^l(x) - \mu^l}{s^l} \odot \gamma_l + \beta_l\right) \quad, \tag{13}$$

where $\odot$ denotes the elementwise (Hadamard) product, $\mu^l \in \mathbb{R}^{N_l}$ and $(s^l)^2 := s^l \odot s^l \in \mathbb{R}^{N_l}$ denote the mean and variance of the pre-activation layers with respect to the data distribution,

$$\mu^l := \mathbb{E}_x\big[z^l(x)\big] \ , \tag{14}$$

$$(s^l)^2 := \mathbb{E}_x\big[(z^l(x) - \mu^l)^2\big] \ . \tag{15}$$

The weights and biases are drawn from the same distributions as in the standard, no BatchNorm, case. In addition, we now have the BatchNorm parameters $\gamma^{l+1}, \beta^{l+1} \in \mathbb{R}^{N_{l+1}}$ which are assumed to be non-random for simplicity,

$$\gamma_l[i] = \gamma_l \ , \qquad \beta_l[i] = 0 \ . \tag{16}$$

### 3.1.1 ORDER PARAMETERS AND THEIR RECURSIONS

To investigate the spectral properties of the FIM, we define the following order parameters,

$$q^l := \frac{1}{N_l} \mathbb{E}_{x,\theta}\big[\|z^l(x)\|^2\big] \ , \qquad\qquad \hat{q}^l := \frac{1}{N_l} \mathbb{E}_{x,\theta}\big[\|h^l(x)\|^2\big] \ , \tag{17}$$

$$q^l_{xy} := \frac{1}{N_l} \mathbb{E}_{x,y,\theta}\big[\langle z^l(x), z^l(y)\rangle\big] \ , \qquad\qquad \hat{q}^l_{xy} := \frac{1}{N_l} \mathbb{E}_{x,y,\theta}\big[\langle h^l(x), h^l(y)\rangle\big] \ , \tag{18}$$

$$\tilde{q}^l := \mathbb{E}_{x,\theta}\big[\|\delta^l(x)\|^2\big] \ , \qquad\qquad \tilde{q}^l_{xy} := \mathbb{E}_{x,y,\theta}\big[\langle \delta^l(x), \delta^l(y)\rangle\big] \ , \tag{19}$$

where $\| \bullet \|$ denotes the Euclidean norm and $\delta^l(x) := \frac{\partial f_\theta}{\partial z^l}(x)$. Here we assume that the data $x$ are drawn i.i.d. from a distribution with mean 0 and variance 1, and also that the last layer is linear for classification. We then have the base cases: $\hat{q}^0 = 0$, $\hat{q}^0_{xy} = 1$, $\tilde{q}^L = \tilde{q}^L_{xy} = 1$. The order parameters in the absence of BatchNorm satisfy the following recursions derived in Karakida et al. (2018),

$$q^l = \sigma_{\mathrm{b}}^2 + \sigma_{\mathrm{w}}^2 \hat{q}^{l-1} \ , \qquad\qquad \hat{q}^l = \frac{q^l}{2} \ , \tag{20}$$

$$q^l_{xy} = \sigma_{\mathrm{b}}^2 + \sigma_{\mathrm{w}}^2 \hat{q}^{l-1}_{xy} \ , \qquad\qquad \hat{q}^l_{xy} = \frac{q^l}{2\pi}\left(\sqrt{1 - c_{xy}^2} + \frac{c_{xy}\pi}{2} + c_{xy}\sin^{-1} c_{xy}\right) \ , \tag{21}$$

$$\tilde{q}^l = \frac{\sigma_{\mathrm{w}}^2}{2}\tilde{q}^{l+1} \ , \qquad\qquad \tilde{q}^l_{xy} = \frac{\sigma_{\mathrm{w}}^2 \tilde{q}^{l+1}_{xy}}{2\pi}\left(\frac{\pi}{2} + \sin^{-1} c_{xy}\right) \ , \tag{22}$$

where $c_{xy} := q^l_{xy}/q^l$. In the case of batch normalization we find the following recursions, which are derived in the Supplementary Material,

$$q^l = \sigma_{\mathrm{b}}^2 + \sigma_{\mathrm{w}}^2 \hat{q}^{l-1} \ , \qquad\qquad \hat{q}^l = \frac{\gamma_l^2}{2} \ , \tag{23}$$

$$q^l_{xy} = \sigma_{\mathrm{b}}^2 + \sigma_{\mathrm{w}}^2 \hat{q}^{l-1}_{xy} \ , \qquad\qquad \hat{q}^l_{xy} = \frac{\gamma_l^2}{2\pi} \ , \tag{24}$$

$$\tilde{q}^l = \frac{\gamma_l^2 \sigma_{\mathrm{w}}^2}{2}\frac{\tilde{q}^{l+1}}{q^l} \ , \qquad\qquad \tilde{q}^l_{xy} = \frac{\gamma_l^2 \sigma_{\mathrm{w}}^2}{4}\frac{\tilde{q}^{l+1}_{xy}}{q^l} \ . \tag{25}$$

### 3.2 CONVOLUTIONAL LAYER

The mean field theory of convolutional layer was first studied by Xiao et al. (2018). In this paper, the results of the preceding section also apply to structured affine transformations including convolutional layers. Let $\mathcal{K}_l$ denote the set of allowable spatial locations of the the $l$th layer feature map and let $\mathcal{F}_{l+1}$ index the sites of the convolutional kernel applied to that layer. Let $C_l$ denote the number of input channels. The output of a general convolutional layer is of the form,

$$z^{l+1}_\alpha(x) = \sum_{\beta \in \mathcal{F}_{l+1}} W^{l+1}_\beta h^l_{\alpha+\beta}(x) + b^{l+1} \ , \tag{26}$$

where $\alpha \in \mathcal{K}_{l+1}$, $W^{l+1}_\beta \in \mathbb{R}^{C_{l+1} \times C_l}$ and $b^{l+1} \in \mathbb{R}^{C_{l+1}}$. The weights and biases are now distributed as

$$[W^{l+1}_\alpha]_{ij} \sim N(0, \sigma_{\mathrm{w}}^2/N_l) \ , \qquad b^{l+1}_i \sim N(0, \sigma_{\mathrm{b}}^2) \ . \tag{27}$$

where now $N_l := C_l|\mathcal{F}_{l+1}|$. As in the fully connected case, we consider convolutional layers with both vanilla activation functions of the form $h_\alpha^l(x) := \sigma(z_\alpha^l(x))$ as well as batch normalized convolutional layers, for which the input activations satisfy the recursive identity,

$$h_\alpha^l(x) := \sigma\left(\frac{z_\alpha^l(x) - \mu_\alpha^l}{s_\alpha^l} \odot \gamma_l + \beta_l\right) \quad, \tag{28}$$

### 3.2.1 Order Parameters and Their recursions

Similar to the definitions for fully connected layer, we define the following set of order parameters:

$$q^l := \frac{1}{C_l}\mathbb{E}_\alpha \mathbb{E}_{x,\theta}\left[\|z_\alpha^l(x)\|^2\right] \quad, \qquad\qquad \hat{q}^l := \frac{1}{C_l}\mathbb{E}_\alpha \mathbb{E}_{x,\theta}\left[\|h_\alpha^l(x)\|^2\right] \quad, \tag{29}$$

$$q_{xy}^l := \frac{1}{C_l}\mathbb{E}_\alpha \mathbb{E}_{x,y,\theta}\left[\langle z_\alpha^l(x), z_\alpha^l(y)\rangle\right] \quad, \qquad \hat{q}_{xy}^l := \frac{1}{C_l}\mathbb{E}_\alpha \mathbb{E}_{x,y,\theta}\left[\langle h_\alpha^l(x), h_\alpha^l(y)\rangle\right] \quad, \tag{30}$$

$$q_{\alpha\beta,xy}^l := \frac{1}{C_l}\mathbb{E}_{\alpha\neq\beta}\left[\mathbb{E}_{x,y,\theta}\langle z_\alpha^l(x), z_\beta^l(y)\rangle\right] \quad, \quad \hat{q}_{\alpha\beta,xy}^l := \frac{1}{C_l}\mathbb{E}_{\alpha\neq\beta}\left[\mathbb{E}_{x,y,\theta}\langle h_\alpha^l(x), h_\beta^l(y)\rangle\right] \quad, \tag{31}$$

$$\tilde{q}^l := \mathbb{E}_\alpha \mathbb{E}_{x,\theta}\left[\|\delta_\alpha^l(x)\|^2\right] \quad, \qquad\qquad \tilde{q}_{xy}^l := \mathbb{E}_\alpha\left[\mathbb{E}_{x,y,\theta}\langle \delta_\alpha^l(x), \delta_\alpha^l(y)\rangle\right] \quad, \tag{32}$$

$$\tilde{q}_{\alpha\beta,xy}^l := \mathbb{E}_{\alpha\neq\beta}\left[\mathbb{E}_{x,y,\theta}\langle \delta_\alpha^l(x), \delta_\beta^l(y)\rangle\right] \quad, \tag{33}$$

where now $\delta_\alpha^l := \partial f_\theta/\partial z_\alpha^l$ in analogy with the fully connected layer. The expectations over $\alpha$ and $\beta$ are with respect to the uniform measure over the set of allowed indices. For a standard convolutional layer without BatchNorm, the order parameters can be shown to satisfy the following recursion relations:

$$q^l = \sigma_\text{b}^2 + \sigma_\text{w}^2\hat{q}_{l-1} \quad, \qquad\qquad \hat{q}^l = \frac{q_l}{2} \quad, \tag{34}$$

$$q_{xy}^l = \sigma_\text{b}^2 + \sigma_\text{w}^2\hat{q}_{xy}^{l-1} \quad, \qquad \hat{q}_{xy}^l = \frac{q^l}{2\pi}\left(\sqrt{1-c_{xy}^2} + \frac{c_{xy}\pi}{2} + c_{xy}\sin^{-1}c_{xy}\right) \quad, \tag{35}$$

$$q_{\alpha\beta,xy}^l = \sigma_\text{b}^2 + \sigma_\text{w}^2\hat{q}_{\alpha\beta,xy}^{l-1} \quad, \qquad \hat{q}_{\alpha\beta,xy}^l = \frac{q^l}{2\pi}\left(\sqrt{1-c_{\alpha\beta}^2} + \frac{c_{\alpha\beta}\pi}{2} + c_{\alpha\beta}\sin^{-1}c_{\alpha\beta}\right) \quad, \tag{36}$$

$$\tilde{q}^l = \frac{\sigma_\text{w}^2}{2}\tilde{q}^{l+1} \quad, \qquad\qquad \tilde{q}_{xy}^l = \frac{\sigma_\text{w}^2\tilde{q}_{xy}^{l+1}}{2\pi}\left(\frac{\pi}{2} + \sin^{-1}c_{xy}\right) \quad, \tag{37}$$

$$\tilde{q}_{\alpha\beta,xy}^l = \frac{\sigma_\text{w}^2\tilde{q}_{\alpha\beta,xy}^{l+1}}{2\pi}\left(\frac{\pi}{2} + \sin^{-1}c_{\alpha\beta}\right) \quad. \tag{38}$$

In the case of convolutional layers with BatchNorm, the following recursions hold:

$$q^l = \sigma_\text{b}^2 + \sigma_\text{w}^2\hat{q}_{l-1} \quad, \qquad\qquad\qquad \hat{q}^l = \frac{\gamma_l^2}{2} \quad, \tag{39}$$

$$q_{xy}^l = \sigma_\text{b}^2 + \sigma_\text{w}^2\hat{q}_{xy}^{l-1} \quad, \qquad\qquad\qquad \hat{q}_{xy}^l = \frac{\gamma_l^2}{2\pi} \quad, \tag{40}$$

$$q_{\alpha\beta,xy}^l = \sigma_\text{b}^2 + \sigma_\text{w}^2\hat{q}_{\alpha\beta,xy}^{l-1} \quad, \qquad\qquad \hat{q}_{\alpha\beta,xy}^l = \frac{\gamma_l^2}{2\pi} \quad, \tag{41}$$

$$\tilde{q}^l = \frac{\gamma_l^2\sigma_\text{w}^2}{2}\frac{\tilde{q}^{l+1}}{q^l} \quad, \qquad\qquad\qquad \tilde{q}_{xy}^l = \frac{\gamma_l^2\sigma_\text{w}^2}{4}\frac{\tilde{q}_{xy}^{l+1}}{q^l} \quad, \tag{42}$$

$$\tilde{q}_{\alpha\beta,xy}^l = \frac{\gamma_l^2\sigma_\text{w}^2}{4}\frac{\tilde{q}_{\alpha\beta,xy}^{l+1}}{q^l} \quad, \tag{43}$$

where $c_{xy} := q_{xy}^l/q^l$ and $c_{\alpha\beta} := q_{\alpha\beta,xy}^l/q^l$. The derivations of the recursion relations for both vanilla and batch-normalized convolutional layers are deferred to the Supplementary Material.

### 3.3 EIGENVALUE BOUND AND THERMODYNAMIC VARIABLES

The order parameters derived in the previous section are useful because they allow us to gain information about the maximal eigenvalue $\bar{\lambda}_{\max}$ of the FIM. We derived a generalization of (Karakida et al., 2018, Theorem 6) to allow for the inclusion of batch normalization and convolutional layers. In particular, we obtain a lower bound on the maximal eigenvalue $\bar{\lambda}_{\max}$ in terms of the previously introduced order parameters which satisfy the stated recursion relations in the mean-field approximation.

**Claim 3.1.** *If the layer dimension $N_l$ of the fully connected layers and the number of channels $C_l$ of the convolutional layers satisfy $N_l \gg 1$ and $C_l \gg 1$ for $0 < l < L$, we have,*

$$\bar{\lambda}_{\max} \geq \sum_{l \in [L]} f_l \ , \tag{44}$$

*where $f_l = N_{l-1} \hat{q}_{xy}^{l-1} \tilde{q}_{xy}^l$ for fully connected layers and*

$$f_l = N_{l-1} \left[ (|\mathcal{K}_l| - 1)\tilde{q}_{\alpha\beta,xy}^l + \tilde{q}_{xy}^l \right] \left[ (|\mathcal{K}_l| - 1)\hat{q}_{\alpha\beta,xy}^{l-1} + \hat{q}_{xy}^{l-1} \right] \ , \tag{45}$$

*for convolutional layers, where recall $N_{l-1} = C_{l-1}|\mathcal{F}_l|$. The index sets $\mathcal{F}_l$ and $\mathcal{K}_l$ are defined in section 3.2. The order parameters are defined in the previous subsection.*

Now we are ready to calculate the lower bound on $\bar{\lambda}_{\max}$ for a given model architecture by calculating the order parameters using their recursions. In the next section, we will focus on the numerical analysis of these recursion relations as well as present experiments that support our calculation.

## 4 NUMERICAL ANALYSIS AND EXPERIMENTS

In order to understand the effect of BatchNorm on the loss landscape, we theoretically compute $\bar{\lambda}_{\max}$ as a function of the BatchNorm parameter $\gamma$, for both fully connected and convolutional architectures (Fig. 1) with and without BatchNorm. For $\gamma \lesssim 3$ (typical for deep network initialization (Ioffe & Szegedy, 2015)) BatchNorm significantly reduces $\bar{\lambda}_{\max}$ compared to the vanilla networks. As a direct consequence of this, the theory predicts that batch normalized networks can be trained using significantly higher learning rates than their vanilla counterparts.

We tested the above theoretical prediction by training the same architectures on MNIST and CIFAR-10 datasets, for different values of $\eta$ and $\gamma$, starting from randomly initialized networks with same variances employed in the mean-field theory calculations. As shown in Fig. 2, the $(\gamma, \log_{10} \eta)$-plane clearly partitions into distinct phases characterized by convergent and non-convergent optimization dynamics, and our theoretically predicted upper bound $\eta_*$ closely agrees with the experimentally determined phase boundary. The experiment of vanilla network is shown in 6.4 as a baseline.

In addition to the striking match between our theoretical prediction and the experimentally determined phase boundaries, the experimental results also suggest a tendency for smaller $\gamma$-initiations to produce lower values of test loss after a fixed number of epochs, i.e. faster convergence. We leave detailed investigation of this initialization scheme to future work. Also, dark strips can be observed in the heatmaps indicate the optimal learning rates for optimization, which is around $\eta_*/2$ and consistent with LeCun et al. (2012) in the quadratic approximation to the loss. Our analysis also suggests that small $\gamma$ initialization benefits the convergence of training. Additional experiments supporting this intuition can be found in Section 6.5 of Supplementary Material.

The architectural design for our experiments is as follows. In the fully connected architecture, we choose $L = 4$ layers with $N_l = 1000$ hidden units per layer except the final (linear) layer which has $N_L = 10$ outputs. Batch normalization is applied after each linear operation except for the final linear output layer. The convolutional network has a similar structure with $L = 4$ layers. The first three are convolutional layers with filter size 3, stride 2, and number of channels $C_1 = 30$, $C_2 = 60$, $C_3 = 90$. The final layer is a fully connected output layer to perform classification. The other architectural/optimization hyperparameters were chosen to be $\sigma_w^2 = 2$, $\sigma_b^2 = 0.5$, $\beta = 0$ and $\mu = 0.9$. Momentum $\mu$ here was set to be 0.9 to match the value frequently used in practice, which only affects the dependency of $\eta_*$ on FIM.

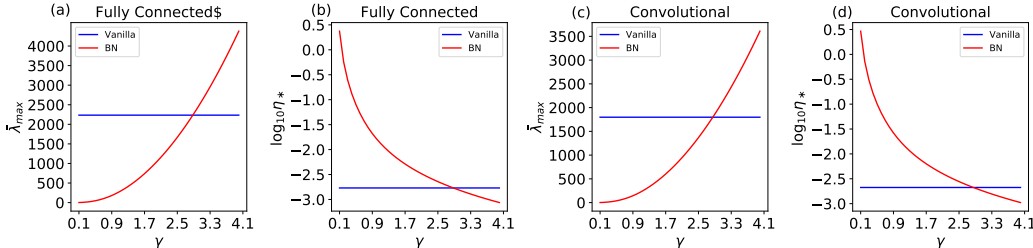

Figure 1: The maximum eigenvalue $\bar{\lambda}_{\max}$ and associated critical learning rate $\eta_*$ for vanilla (blue) and BatchNorm networks (red) as a function of the BatchNorm parameter $\gamma$ for different choices of architecture (fully-connected and convolutional), calculated by theory. (a, c) shows the flattening effect of BatchNorm on the loss function for a wide range of hyperparameters and (b, d) further show that for sufficiently small $\gamma$ BatchNorm enables optimization with much higher learning rate than vanilla networks.

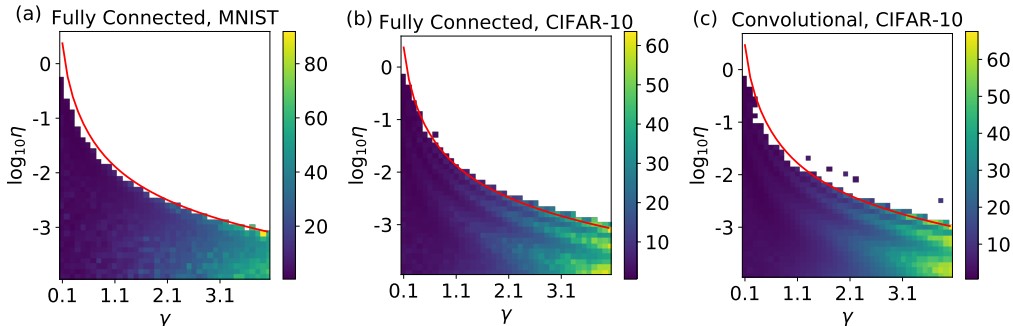

Figure 2: Heatmaps showing test loss as a function of $(\log_{10}\eta, \gamma)$ after 5 epochs of training for different choices of dataset and architecture. Results were obtained by averaging 5 random restarts. The white region indicates parameter explosion for at least one of the runs. The red line shows the theoretical prediction for the maximal learning rate $\eta_*$. The dark band on the heatmaps for CIFAR-10 approximately tracks the optimal learning rate $\eta_*/2$ in the quadratic approximation to the loss. Note the log scale for the learning rate, so the theory matches the experiments over three orders of magnitude for $\eta$.

## 5  CONCLUSION AND FUTURE WORK

In this paper, we studied the impact of BatchNorm on the loss surface of multi-layer neural networks and its implication for training dynamics. By developing recursion relations for the relevant order parameters, the maximum eigenvalue of the Fisher Information matrix $\bar{\lambda}_{\max}$ can be estimated and related to the maximal learning rate. The theory correctly predicts that adding BatchNorm with small $\gamma$ allows the training algorithm to exploit much larger learning rates, which speeds up convergence. The experiments also suggest that using a smaller $\gamma$ results in a lower test loss for a fixed number of training epochs. This suggests that initialization with smaller $\gamma$ may help the optimization process in deep learning models, which will be interesting for future study.

The close agreement between theoretical predictions and the experimentally determined phase boundaries strongly supports the validity of our analysis, despite the non-rigorous nature of the derivations. Although similar approaches have been used in other work (Poole et al., 2016; Schoenholz et al., 2017; Yang & Schoenholz, 2017; Xiao et al., 2018; Karakida et al., 2018), we hope that future work will place these results on a firmer mathematical footing. Furthermore, our BatchNorm analysis is not limited to the convolutional and fully-connected architectures we considered in this paper and can be extended to arbitrary feedforward architectures such as ResNets.

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

# 6 SUPPLEMENTARY MATERIAL

This section provides non-rigorous derivations of the order parameters, their recursions, and the associated eigenvalue bound. Despite the non-rigorous nature of these calculations, we remark that similar reasoning has been successfully used in a number of related works on mean-field theory, demonstrating impressive agreement with experiments (Poole et al., 2016; Schoenholz et al., 2017; Yang & Schoenholz, 2017; Xiao et al., 2018; Karakida et al., 2018).

## 6.1 RECURSIONS FOR FULLY CONNECTED LAYERS

**Claim 6.1.** *The forward recursions for $0 \leq l \leq L-1$ are $q^{l+1} = \sigma_b^2 + \sigma_w^2 \hat{q}^l$ and $q_{xy}^{l+1} = \sigma_b^2 + \sigma_w^2 \hat{q}_{xy}^l$ where $\hat{q}^l = \gamma_l^2/2$ and $\hat{q}_{xy}^l = \gamma_l^2/(2\pi)$ for $l \in [L-1]$ whereas $\hat{q}^0 = 1$ and $\hat{q}_{xy}^0 = 0$.*

*Derivation.* In general, we have

$$\frac{1}{N_{l+1}} \mathop{\mathbb{E}}_{\theta} \langle z^{l+1}(x), z^{l+1}(y) \rangle = \sigma_b^2 + \frac{\sigma_w^2}{N_l} \langle h_\theta^l(x), h_\theta^l(y) \rangle \ . \tag{46}$$

Thus, setting $l = 0$ we obtain $q^1 = \sigma_w^2 + \sigma_b^2$ if we assume $\mathop{\mathbb{E}}_x \|x\|^2 = N_0$, and $q_{xy}^1 = \sigma_b^2$ since $\mathop{\mathbb{E}}_{x,y} \langle x, y \rangle = \langle \mathbb{E} x, \mathbb{E} y \rangle = 0$.

Recall (for $l > 0$) $z^{l+1}(x) = W^{l+1} \sigma_l(u^l(x) \odot \gamma_l) + b^{l+1}$ where $u^l(x) = [z^l(x) - \mu^l]/s^l$ so

$$\frac{1}{N_{l+1}} \mathop{\mathbb{E}}_{\theta} \langle z^{l+1}(x), z^{l+1}(y) \rangle = \sigma_b^2 + \frac{\sigma_w^2}{N_l} \langle \sigma_l(u^l(x) \odot \gamma_l), \sigma_l(u^l(y) \odot \gamma_l) \rangle \ . \tag{47}$$

Therefore, setting $x = y$ and taking expectation values over $x$ gives,

$$q^{l+1} := \frac{1}{N_{l+1}} \mathop{\mathbb{E}}_{x,\theta} \|z^{l+1}(x)\|^2 \ , \tag{48}$$

$$= \sigma_b^2 + \frac{\sigma_w^2}{N_l} \mathop{\mathbb{E}}_{x,\theta} \|h^l(x)\|^2 \ , \tag{49}$$

$$= \sigma_b^2 + \sigma_w^2 \hat{q}^l \ , \tag{50}$$

$$\hat{q}^l := \frac{1}{N_l} \mathop{\mathbb{E}}_{x,\theta} \|h^l(x)\|^2 \ , \tag{51}$$

$$= \sigma_w^2 \mathop{\mathbb{E}}_{x,\theta} \sigma_l^2(\gamma_l u^l(x)[1]) \ , \tag{52}$$

$$\simeq \sigma_w^2 \int Dz \, \sigma_l^2(\gamma_l z) \ , \tag{53}$$

$$= \frac{\gamma_l^2}{2} \ , \tag{54}$$

where we have approximated each component of the random vector $u^l(x)$ as a standard Gaussian.

Similarly, taking expectations over $x, y$ gives

$$q_{xy}^{l+1} = \sigma_b^2 + \sigma_w^2 \mathop{\mathbb{E}}_{x,y,\theta} \sigma_l(u^l(x)[1]\,\gamma_l) \sigma_l(u^l(y)[1]\,\gamma_l) \ . \tag{55}$$

Consider the approximation in which the random pair $(u^l(x)[1], u^l(y)[1])$ is Gaussian distributed with zero mean and covariance,

$$\Sigma^l := \begin{pmatrix} \Sigma_{xx}^l & \Sigma_{xy}^l \\ \Sigma_{xy}^l & \Sigma_{yy}^l \end{pmatrix} := \begin{pmatrix} \mathbb{E} \, u^l(x)[1]^2 & \mathbb{E} \, u^l(x)[1] u^l(y)[1] \\ \mathbb{E} \, u^l(x)[1] u^l(y)[1] & \mathbb{E} \, u^l(y)[1]^2 \end{pmatrix} \ . \tag{56}$$

Then the recursion becomes

$$q_{xy}^{l+1} \simeq \sigma_b^2 + \sigma_w^2 \int Dz_1 \, Dz_2 \, \sigma_l\left(\sqrt{\Sigma_{xx}^l} \, z_1 \gamma_l\right) \sigma_l\left[\sqrt{\Sigma_{yy}^l} \left(c_{xy}^l \, z_1 + \sqrt{1 - (c_{xy}^l)^2} \, z_2\right) \gamma_l\right] \ , \tag{57}$$

where $c_{xy}^l := \Sigma_{xy}^l / \sqrt{\Sigma_{xx}^l \Sigma_{yy}^l}$. Observe that by independence of $x$ and $y$,

$$\mathbb{E}_{x,y}\left[u_l(x)u_l(y)\right] = \mathbb{E}_{x,y}\left[\frac{z_l(x)z_l(y) + \mu_l^2 - \mu_l\big(z_l(x) + z_l(y)\big)}{s_l^2}\right] , \tag{58}$$

$$= \frac{\mathbb{E}_x z_l(x)\mathbb{E}_y z_l(y) + \mu_l^2 - \mu_l\big(\mathbb{E}_x z_l(x) + \mathbb{E}_y z_l(y)\big)}{s_l^2} , \tag{59}$$

$$= 0 . \tag{60}$$

Thus $\Sigma_{xy}^l = 0$ and consequently,

$$q_{xy}^{l+1} = \sigma_b^2 + \sigma_w^2 \hat{q}_{xy}^l , \tag{61}$$

$$\hat{q}_{xy}^l = \frac{\gamma_l^2}{2\pi} . \tag{62}$$

$\square$

The derivation here assumes an infinitely large dataset. For limited dataset size m, an error is introduced from the non-zero ratio of $m/m_{x\neq y}$ where $m_{x\neq y}$ denotes the total number of sample pairs $(x, y)$ where $x \neq y$. We have $m_{x\neq y} = m^2 - m$ and therefore the error is $O(1/m)$ which is negligible for most of the frequently-used datasets.

**Claim 6.2.** *The backward recursions for $l \in [1, L-1]$ are $\tilde{q}^l = \frac{\gamma_l^2 \sigma_w^2}{2} \frac{\tilde{q}^{l+1}}{q^l}$ and $\tilde{q}_{xy}^l = \frac{\gamma_l^2 \sigma_w^2}{4} \frac{\tilde{q}_{xy}^{l+1}}{q^l}$ with base cases $\tilde{q}^L = \tilde{q}_{xy}^L = 1$.*

*Derivation.* For each layer $l \in [L]$ and for each unit $i \in [N_l]$ define $\delta_l[i] \in \mathbb{R}^{N_L}$ by

$$\delta^l[i] := \frac{\partial h_\theta^L}{\partial z^l[i]} . \tag{63}$$

In particular, since we assume linear output $\delta^L[i] = e_i$ and thus

$$\tilde{q}^L = \mathbb{E}_{x,\theta}\|\delta^L[\cdot]\|^2 = \mathbf{1} , \tag{64}$$

where $\mathbf{1} \in \mathbb{R}^{N_L}$ is the vector of ones. For ease of presentation and without loss of generality we restrict to the first component $\tilde{q}^L[1] = 1$ and abuse notation by writing $\tilde{q}^L = 1$. For $l < L$ we have by the chain rule,

$$\delta^l[i] = \sum_{k\in[N_{l+1}]} \frac{\partial z^{l+1}[k]}{\partial z^l[i]} \delta^{l+1}[k] , \tag{65}$$

$$= \frac{\gamma_l[i]}{s^l[i]}\sigma_l'\big(u^l[i]\,\gamma_l[i]\big) \sum_{k\in[N_{l+1}]} [W^{l+1}]_{ki}\delta^{l+1}[k] . \tag{66}$$

Hence,

$$\delta^l(x)[i]\delta^l(y) = \frac{\gamma_l[i]^2}{s^l[i]^2}\sigma_l'\big(u^l(x)[i]\,\gamma_l[i]\big)\,\sigma_l'\big(u^l(y)[i]\,\gamma_l[i]\big) \sum_{k,k'\in[N_{l+1}]} [W^{l+1}]_{ki}[W^{l+1}]_{k'i}\delta^{l+1}[k]\delta^{l+1}[k'] . \tag{67}$$

Following the usual assumption that the back-propagated gradient is independent of the forward signal we obtain,

$$\mathbb{E}_\theta\langle\delta^l(x), \delta^l(y)\rangle = \frac{\sigma_w^2}{N_l}\sum_{i\in[N_l]}\mathbb{E}_\theta\left[\frac{\gamma_l[i]^2}{s^l[i]^2}\sigma_l'\big(u^l(x)[i]\,\gamma_l[i]\big)\,\sigma_l'\big(u^l(y)[i]\,\gamma_l[i]\big)\right]\mathbb{E}_\theta\langle\delta^{l+1}(x), \delta^{l+1}(y)\rangle \tag{68}$$

Setting $x = y$ and taking expectation values over $x$ we obtain,

$$\tilde{q}_l := \mathop{\mathbb{E}}_{x,\theta} \|\delta^l(x)\|^2 \ , \tag{69}$$

$$\simeq \frac{\tilde{q}_{l+1}}{q_l} \gamma_l^2 \sigma_{\mathrm{w}}^2 \int Dz\, \sigma_l'(\gamma_l z)^2 \ , \tag{70}$$

$$= \frac{\gamma_l^2 \sigma_{\mathrm{w}}^2}{2} \frac{\tilde{q}_{l+1}}{q_l} \ . \tag{71}$$

Similarly, taking expectation values over $x, y$ we obtain,

$$\tilde{q}_{xy}^l = \mathop{\mathbb{E}}_{x,y,\theta} \langle \delta^l(x), \delta^l(y) \rangle \ , \tag{72}$$

$$= \sigma_{\mathrm{w}}^2 \gamma_l^2 \frac{\tilde{q}_{xy}^{l+1}}{q^l} \int Dz_1 Dz_2 \sigma_l'\Big(\sqrt{\Sigma_{xx}^l} z_1 \gamma_l\Big) \sigma_l'\Big[\sqrt{\Sigma_{yy}^l} \Big(c_{xy}^l z_1 + \sqrt{1 - (c_{xy}^l)^2}\, z_2\Big) \gamma_l\Big] \ , \tag{73}$$

$$= \frac{\sigma_{\mathrm{w}}^2 \gamma_l^2}{4} \frac{\tilde{q}_{xy}^{l+1}}{q^l} \ . \tag{74}$$

$\square$

## 6.2 Recursions for convolutional layers

Recall that the output of a general convolutional layer is of the form,

$$z_\alpha^{l+1}(x) = \sum_{\beta \in \mathcal{F}_{l+1}} W_\beta^{l+1} h_{\alpha+\beta}^l(x) + b^{l+1} \ , \tag{75}$$

where $\alpha \in \mathcal{K}_{l+1}$. As a concrete example, consider CIFAR-10 input of dimension $32 \times 32 \times 3$ which is mapped by a convolutional layer with $3 \times 3$ kernels, stride 2 and 20 output channels and no padding. Then $C_0 = 3$, $C_1 = 20$, $|\mathcal{K}_0| = 1024$ and $|\mathcal{F}_1| = 9$ and $|\mathcal{K}_1| = 225$.

We begin by deriving some useful identities for convolutional layers, before specializing to the batch-normalized and vanilla networks. For each channel $i \in [C_l]$, we have,

$$\mathop{\mathbb{E}}_\theta z_\alpha^{l+1}(x)[i] z_\beta^{l+1}(y)[i] = \sigma_{\mathrm{b}}^2 + \mathop{\mathbb{E}}_\theta \sum_{\substack{(\beta_1,j_1) \in \mathcal{F}_{l+1} \times [C_l] \\ (\beta_2,j_2) \in \mathcal{F}_{l+1} \times [C_l]}} [W_{\beta_1}^{l+1}]_{ij_1} [W_{\beta_2}^{l+1}]_{ij_2} h_{\alpha+\beta_1}^l(x)[j_1] h_{\beta+\beta_2}^l(y)[j_2] \ ,$$

$$= \sigma_{\mathrm{b}}^2 + \frac{\sigma_{\mathrm{w}}^2}{N_l} \sum_{\delta \in \mathcal{F}_{l+1}} \mathop{\mathbb{E}}_\theta \langle h_{\alpha+\delta}^l(x), h_{\beta+\delta}^l(y) \rangle \ . \tag{76}$$

Hence,

$$\frac{1}{C_{l+1}} \mathop{\mathbb{E}}_\theta \langle z_\alpha^{l+1}(x), z_\beta^{l+1}(y) \rangle = \sigma_{\mathrm{b}}^2 + \frac{\sigma_{\mathrm{w}}^2}{N_l} \sum_{\delta \in \mathcal{F}_{l+1}} \mathop{\mathbb{E}}_\theta \langle h_{\alpha+\delta}^l(x), h_{\beta+\delta}^l(y) \rangle \ , \tag{77}$$

where $N_l := C_l |\mathcal{F}_{l+1}|$.

Moreover, let us introduce the following shorthand,

$$\mathop{\mathbb{E}}_\alpha \|h_\alpha^l(x)\|^2 := \frac{1}{|\mathcal{K}_l|} \sum_{\alpha \in \mathcal{K}_l} \|h_\alpha^l(x)\|^2 \ , \tag{78}$$

$$\mathop{\mathbb{E}}_\alpha \langle h_\alpha^l(x), h_\alpha^l(y) \rangle := \frac{1}{|\mathcal{K}_l|} \sum_{\alpha \in \mathcal{K}_l} \langle h_\alpha^l(x), h_\alpha^l(y) \rangle \ , \tag{79}$$

$$\mathop{\mathbb{E}}_{\alpha \neq \beta} \langle h_\alpha^l(x), h_\beta^l(y) \rangle := \frac{1}{|\mathcal{K}_l|(|\mathcal{K}_l| - 1)} \sum_{\alpha,\beta \in \mathcal{K}_l \times \mathcal{K}_l : \alpha \neq \beta} \langle h_\alpha^l(x), h_\beta^l(y) \rangle \tag{80}$$

Then we can write the recursion relations as,

$$q^{l+1} := \frac{1}{C_{l+1}} \mathop{\mathbb{E}}_{\alpha} \mathop{\mathbb{E}}_{x,\theta} \|z_\alpha^{l+1}(x)\|^2 \ , \tag{81}$$

$$= \mathop{\mathbb{E}}_{x} \left[ \sigma_{\mathrm{b}}^2 + \frac{\sigma_{\mathrm{w}}^2}{N_l} \sum_{\delta \in \mathcal{F}_{l+1}} \mathop{\mathbb{E}}_{\theta} \mathop{\mathbb{E}}_{\alpha} \|h_{\alpha+\delta}^l(x)\|^2 \right] \ , \tag{82}$$

$$= \sigma_{\mathrm{b}}^2 + \frac{\sigma_{\mathrm{w}}^2}{C_l} \mathop{\mathbb{E}}_{x,\theta} \mathop{\mathbb{E}}_{\alpha} \|h_\alpha^l(x)\|^2 \ , \tag{83}$$

$$= \sigma_{\mathrm{b}}^2 + \sigma_{\mathrm{w}}^2 \hat{q}^l \ , \tag{84}$$

$$\hat{q}^l := \frac{1}{C_l} \mathop{\mathbb{E}}_{x,\theta} \mathop{\mathbb{E}}_{\alpha} \|h_\alpha^l(x)\|^2 \ . \tag{85}$$

Furthermore we have,

$$q_{xy}^{l+1} = \frac{1}{C_{l+1}} \mathop{\mathbb{E}}_{\alpha} \mathop{\mathbb{E}}_{x,y,\theta} \langle z_\alpha^{l+1}(x), z_\alpha^{l+1}(y) \rangle \ , \tag{86}$$

$$= \mathop{\mathbb{E}}_{x,y} \left[ \sigma_{\mathrm{b}}^2 + \frac{\sigma_{\mathrm{w}}^2}{N_l} \sum_{\delta \in \mathcal{F}_{l+1}} \mathop{\mathbb{E}}_{\theta} \mathop{\mathbb{E}}_{\alpha} \langle h_{\alpha+\delta}^l(x), h_{\alpha+\delta}^l(y) \rangle \right] \ , \tag{87}$$

$$= \sigma_{\mathrm{b}}^2 + \sigma_{\mathrm{w}}^2 \hat{q}_{xy}^l \ , \tag{88}$$

$$\hat{q}_{xy}^l := \frac{1}{C_l} \mathop{\mathbb{E}}_{\alpha} \mathop{\mathbb{E}}_{x,y,\theta} \langle h_\alpha^l(x), h_\alpha^l(y) \rangle \ . \tag{89}$$

Finally, we have,

$$q_{\alpha\beta,xy}^{l+1} := \mathop{\mathbb{E}}_{\alpha \neq \beta} \left[ \mathop{\mathbb{E}}_{x,y,\theta} \langle z_\alpha^{l+1}(x), z_\beta^{l+1}(y) \rangle \right] \ , \tag{90}$$

$$= \mathop{\mathbb{E}}_{x,y} \left[ \sigma_{\mathrm{b}}^2 + \frac{\sigma_{\mathrm{w}}^2}{N_l} \sum_{\delta \in \mathcal{F}_{l+1}} \mathop{\mathbb{E}}_{\theta} \mathop{\mathbb{E}}_{\alpha \neq \beta} \langle h_{\beta+\delta}^l(x), h_{\alpha+\delta}^l(y) \rangle \right] \ , \tag{91}$$

$$= \mathop{\mathbb{E}}_{x,y} \left[ \sigma_{\mathrm{b}}^2 + \frac{\sigma_{\mathrm{w}}^2}{C_l} \mathop{\mathbb{E}}_{\theta} \mathop{\mathbb{E}}_{\alpha \neq \beta} \langle h_\beta^l(x), h_\alpha^l(y) \rangle \right] \ , \tag{92}$$

$$= \sigma_{\mathrm{b}}^2 + \sigma_{\mathrm{w}}^2 \hat{q}_{\alpha\beta,xy}^l \ , \tag{93}$$

$$\hat{q}_{\alpha\beta,xy}^l := \frac{1}{C_l} \mathop{\mathbb{E}}_{\alpha \neq \beta} \mathop{\mathbb{E}}_{x,y,\theta} \langle h_\alpha^l(x), h_\beta^l(y) \rangle \ , \tag{94}$$

where we used,

$$\sum_{\delta \in \mathcal{F}_{l+1}} \sum_{\alpha,\beta \in \mathcal{K}_l \times \mathcal{K}_l : \alpha \neq \beta} \langle h_{\alpha+\delta}^l(x), h_{\beta+\delta}^l(y) \rangle = \tag{95}$$

$$\sum_{\delta \in \mathcal{F}_{l+1}} \left[ \left\langle \sum_{\alpha \in \mathcal{K}_l} h_{\alpha+\delta}^l(x), \sum_{\beta \in \mathcal{K}_l} h_{\beta+\delta}^l(y) \right\rangle - \sum_{\alpha \in \mathcal{K}_l} \langle h_{\alpha+\delta}^l(x), h_{\alpha+\delta}^l(y) \rangle \right] \tag{96}$$

$$= |\mathcal{F}_{l+1}| \left[ \left\langle \sum_{\alpha \in \mathcal{K}_l} h_\alpha^l(x), \sum_{\beta \in \mathcal{K}_l} h_\beta^l(y) \right\rangle - \sum_{\alpha \in \mathcal{K}_l} \langle h_\alpha^l(x), h_\alpha^l(y) \rangle \right] \tag{97}$$

$$= |\mathcal{F}_{l+1}| \sum_{\alpha,\beta \in \mathcal{K}_l \times \mathcal{K}_l : \alpha \neq \beta} \langle h_\alpha^l(x), h_\beta^l(y) \rangle \ . \tag{98}$$

### 6.2.1 BATCH NORMALIZATION

**Claim 6.3.** *The forward recursions for $0 \leq l \leq L-1$ are $q^{l+1} = \sigma_{\mathrm{b}}^2 + \sigma_{\mathrm{w}}^2 \hat{q}^l$, $q_{xy}^{l+1} = \sigma_{\mathrm{b}}^2 + \sigma_{\mathrm{w}}^2 \hat{q}_{xy}^l$ and $q_{\alpha\beta,xy}^{l+1} = \sigma_{\mathrm{b}}^2 + \sigma_{\mathrm{w}}^2 \hat{q}_{\alpha\beta,xy}^l$ where for each $l \in [L-1]$ we have $\hat{q}^l = \gamma_l^2/2$ and $\hat{q}_{xy}^l = \hat{q}_{\alpha\beta,xy}^l = \gamma_l^2/(2\pi)$.*

*Derivation.* Recalling that $h_\alpha^l(x) = \sigma_l\big(u_\alpha^l(x) \odot \gamma_l\big)$ where $u_\alpha^l(x) := \frac{z_\alpha^l(x) - \mu_\alpha^l}{s_\alpha^l}$ and substituting into equation 85, equation 89 and equation 94 we obtain,

$$\hat{q}^l \simeq \int Dz\, \sigma_l^2(\gamma_l z) \ , \tag{99}$$

$$= \frac{\gamma_l^2}{2} \ , \tag{100}$$

$$\hat{q}_{xy}^l \simeq \int Dz_1\, Dz_2\, \sigma_l(\gamma_l z_1)\sigma_l(\gamma_1 z_2) \ , \tag{101}$$

$$= \frac{\gamma_l^2}{2\pi} \ , \tag{102}$$

$$\hat{q}_{\alpha\beta,xy} \simeq \int Dz_1\, Dz_2\, \sigma_l(\gamma_l z_1)\sigma_l(\gamma_1 z_2) \ , \tag{103}$$

$$= \frac{\gamma_l^2}{2\pi} \ , \tag{104}$$

$\square$

**Claim 6.4.** *The backward recursions are* $\tilde{q}^l = \frac{\sigma_w^2 \gamma_l^2 \tilde{q}^{l+1}}{2q^l}$, $\tilde{q}_{xy}^l = \frac{\sigma_w^2 \gamma_l^2 \tilde{q}_{xy}^{l+1}}{4q^l}$, *and* $\tilde{q}_{\alpha\beta,xy}^l = \frac{\sigma_w^2 \gamma_l^2 \tilde{q}_{\alpha\beta,xy}^{l+1}}{4q^l}$.

In order to derive the backward recursion, define (for each $l \in [L]$, $i \in [C_l]$, $\alpha \in \mathcal{K}_l$)

$$\delta_\alpha^l[i] = \frac{\partial h_\theta^L}{\partial z_\alpha^l[i]} \ . \tag{105}$$

Then by the chain rule,

$$\delta_\alpha^l[i] = \sum_{(\beta,k) \in \mathcal{K}_{l+1} \times [C_{l+1}]} \frac{\partial z_\beta^{l+1}[k]}{\partial z_\alpha^l[i]} \delta_\beta^{l+1}[k] \ . \tag{106}$$

Now,

$$z_\beta^{l+1}[k] = \sum_{(\beta',j) \in \mathcal{F}_{l+1} \times [C_l]} [W_{\beta'}^{l+1}]_{kj}\sigma_l\big(u_{\beta+\beta'}^l[j]\,\gamma_l[j]\big) + b^{l+1}[k] \ , \tag{107}$$

$$\frac{\partial z_\beta^{l+1}[k]}{\partial z_\alpha^l[i]} = \sum_{(\beta',j) \in \mathcal{F}_{l+1} \times [C_l]} [W_{\beta'}^{l+1}]_{kj}\sigma_l'\big(u_{\beta+\beta'}^l[j]\,\gamma_l[j]\big)\frac{\gamma_l[j]}{s_{\beta+\beta'}^l[j]}\delta_{ij}\delta_{\alpha,\beta+\beta'} \ , \tag{108}$$

$$= [W_{\alpha-\beta}^{l+1}]_{ki}\sigma_l'\big(u_\alpha^l[i]\,\gamma_l[i]\big)\frac{\gamma_l[i]}{s_\alpha^l[i]} \ . \tag{109}$$

Thus,

$$\delta_\alpha^l[i] = \frac{\gamma_l[i]}{s_\alpha^l[i]}\sigma_l'\big(u_\alpha^l[i]\,\gamma_l[i]\big)\sum_{(\beta,k) \in \mathcal{F}_{l+1} \times [C_{l+1}]} [W_\beta^{l+1}]_{ki}\delta_{\alpha-\beta}^{l+1}[k] \ . \tag{110}$$

By the distributional assumption on the weights, for each $(\alpha,\beta) \in \mathcal{K}_l \times \mathcal{K}_l$ we have,

$$\underset{\theta}{\mathbb{E}} \sum_{\substack{(\beta_1,k_1) \in \mathcal{F}_{l+1} \times [C_{l+1}] \\ (\beta_2,k_2) \in \mathcal{F}_{l+1} \times [C_{l+1}]}} [W_{\beta_1}^{l+1}]_{k_1 i}[W_{\beta_2}^{l+1}]_{k_2 i}\delta_{\alpha-\beta_1}^{l+1}[k_1]\delta_{\beta-\beta_2}^{l+1}[k_2] = \frac{\sigma_w^2}{N_l}\sum_{\delta \in \mathcal{F}_{l+1}} \underset{\theta}{\mathbb{E}} \langle \delta_{\alpha-\delta}^{l+1}(x), \delta_{\beta-\delta}^{l+1}(y)\rangle \ . \tag{111}$$

Thus, under the usual independence assumptions,

$$\underset{\theta}{\mathbb{E}} \langle \delta_\alpha^l(x), \delta_\beta^l(y)\rangle = \frac{\sigma_w^2}{N_l}\sum_{i \in [C_l]} \underset{\theta}{\mathbb{E}} \left[\frac{\gamma_l[i]^2}{s_\alpha^l[i]s_\beta^l[i]}\sigma_l'\big(u_\alpha^l(x)[i]\,\gamma_l[i]\big)\sigma_l'\big(u_\beta^l(y)[i]\,\gamma_l[i]\big)\right]\sum_{\delta \in \mathcal{F}_{l+1}} \underset{\theta}{\mathbb{E}} \langle \delta_{\alpha-\delta}^{l+1}(x), \delta_{\beta-\delta}^{l+1}(y)\rangle \ . \tag{112}$$

Setting $\alpha = \beta$, $x = y$, averaging over $\alpha$ and taking the expectation value over $x$,

$$\tilde{q}^l := \underset{\alpha}{\mathbb{E}} \underset{x,\theta}{\mathbb{E}} \|\delta_\alpha^l(x)\|^2 \quad , \tag{113}$$

$$= \frac{\sigma_{\mathrm{w}}^2}{N_l} \frac{\gamma_l^2}{2q^l} C_l \underset{x,\theta}{\mathbb{E}} \sum_{\delta \in \mathcal{F}_{l+1}} \underset{\alpha}{\mathbb{E}} \|\delta_{\alpha-\delta}^l(x)\|^2 \quad , \tag{114}$$

$$= \frac{\sigma_{\mathrm{w}}^2}{N_l} \frac{\gamma_l^2}{2q^l} C_l |\mathcal{F}_{l+1}| \underset{\alpha}{\mathbb{E}} \underset{x,\theta}{\mathbb{E}} \|\delta_\alpha^l(x)\|^2 \quad , \tag{115}$$

$$= \frac{\sigma_{\mathrm{w}}^2 \gamma_l^2 \tilde{q}^{l+1}}{2q^l} \quad . \tag{116}$$

Similarly, setting $\alpha = \beta$, averaging over $\alpha$ and taking expectation values over $x, y$ gives

$$\tilde{q}_{xy}^l := \underset{\alpha}{\mathbb{E}} \underset{x,y,\theta}{\mathbb{E}} \langle \delta_\alpha^l(x), \delta_\alpha^l(y) \rangle \quad , \tag{117}$$

$$= \frac{\sigma_{\mathrm{w}}^2 \gamma_l^2 \tilde{q}^{l+1}}{4q^l} \quad . \tag{118}$$

Now,

$$\sum_{\substack{\alpha,\beta \in \mathcal{K}_{l+1} \times \mathcal{K}_{l+1} : \alpha \neq \beta \\ \delta \in \mathcal{F}_{l+1}}} \langle \delta_{\alpha-\delta}^{l+1}(x), \delta_{\beta-\delta}^{l+1}(y) \rangle \tag{119}$$

$$= \sum_{\delta \in \mathcal{F}_{l+1}} \left[ \left\langle \sum_{\alpha \in \mathcal{K}_{l+1}} \delta_{\alpha-\delta}^{l+1}(x), \sum_{\beta \in \mathcal{K}_{l+1}} \delta_{\beta-\delta}^{l+1}(y) \right\rangle - \sum_{\alpha \in \mathcal{K}_{l+1}} \langle \delta_{\alpha-\delta}^{l+1}(x), \delta_{\alpha-\delta}^{l+1}(y) \rangle \right] \quad , \tag{120}$$

$$= |\mathcal{F}_{l+1}| \left[ \left\langle \sum_{\alpha \in \mathcal{K}_{l+1}} \delta_\alpha^{l+1}(x), \sum_{\beta \in \mathcal{K}_l} \delta_\beta^{l+1}(y) \right\rangle - \sum_{\alpha \in \mathcal{K}_{l+1}} \langle \delta_\alpha^{l+1}(x), \delta_\alpha^{l+1}(y) \rangle \right] \quad , \tag{121}$$

$$= |\mathcal{F}_{l+1}| \sum_{\alpha,\beta \in \mathcal{K}_{l+1} \times \mathcal{K}_{l+1} : \alpha \neq \beta} \left\langle \delta_\alpha^{l+1}(x), \delta_\beta^{l+1}(y) \right\rangle \quad . \tag{122}$$

Let us further assume that

$$\frac{1}{|\mathcal{K}_l|(|\mathcal{K}_l| - 1)} \sum_{\alpha,\beta \in \mathcal{K}_l \times \mathcal{K}_l : \alpha \neq \beta} \underset{x,y,\theta}{\mathbb{E}} \left\langle \delta_\alpha^{l+1}(x), \delta_\beta^{l+1}(y) \right\rangle \tag{123}$$

$$= \frac{1}{|\mathcal{K}_{l+1}|(|\mathcal{K}_{l+1}| - 1)} \sum_{\alpha,\beta \in \mathcal{K}_{l+1} \times \mathcal{K}_{l+1} : \alpha \neq \beta} \underset{x,y,\theta}{\mathbb{E}} \left\langle \delta_\alpha^{l+1}(x), \delta_\beta^{l+1}(y) \right\rangle \quad . \tag{124}$$

Thus, taking expectation values over $(x, y)$ and averaging over the allowable indices such that $\alpha \neq \beta$ we obtain,

$$\underset{\alpha \neq \beta}{\mathbb{E}} \left[ \underset{x,y,\theta}{\mathbb{E}} \langle \delta_\alpha^l(x), \delta_\beta^l(y) \rangle \right] = \frac{C_l |\mathcal{F}_{l+1}|}{N_l} \frac{\sigma_{\mathrm{w}}^2 \gamma_l^2}{4q^l} \underset{\alpha \neq \beta}{\mathbb{E}} \left[ \underset{x,y,\theta}{\mathbb{E}} \langle \delta_\alpha^{l+1}(x), \delta_\beta^{l+1}(y) \rangle \right] \quad . \tag{125}$$

It follows that

$$\tilde{q}_{\alpha\beta,xy}^l := \underset{\alpha \neq \beta}{\mathbb{E}} \left[ \underset{x,y,\theta}{\mathbb{E}} \langle \delta_\alpha^l(x), \delta_\beta^l(y) \rangle \right] \quad , \tag{126}$$

$$= \frac{\sigma_{\mathrm{w}}^2 \gamma_l^2 \tilde{q}_{\alpha\beta,xy}^{l+1}}{4q^l} \quad . \tag{127}$$

### 6.2.2 VANILLA CNN

**Claim 6.5.** *The forward recursions are $q^{l+1} = \sigma_{\mathrm{b}}^2 + \sigma_{\mathrm{w}}^2 \hat{q}^l$, $q_{xy}^{l+1} = \sigma_{\mathrm{b}}^2 + \sigma_{\mathrm{w}}^2 \hat{q}_{xy}^l$ and $q_{\alpha\beta,xy}^{l+1} = \sigma_{\mathrm{b}}^2 + \sigma_{\mathrm{w}}^2 \hat{q}_{\alpha\beta,xy}^l$ where*

$$\hat{q}^l = \frac{1}{2} q^l \ , \tag{128}$$

$$\hat{q}_{xy}^l = \frac{q^l}{2\pi} \left[ \sqrt{1 - c_{xy}^2} + \frac{c_{xy}\pi}{2} + c_{xy} \sin^{-1}(c_{xy}) \right] \ , \tag{129}$$

$$\hat{q}_{\alpha\beta,xy}^l = \frac{q^l}{2\pi} \left[ \sqrt{1 - c_{\alpha\beta,xy}^2} + \frac{c_{\alpha\beta,xy}\pi}{2} + c_{\alpha\beta,xy} \sin^{-1}(c_{\alpha\beta,xy}) \right] \ , \tag{130}$$

*and $c_{xy} := q_{xy}^l / q^l$, $c_{\alpha\beta,xy} := q_{\alpha\beta,xy}^l / q^l$.*

*Derivation.* Substituting into equation 85, equation 89 and equation 94 we obtain,

$$\hat{q}^l := \int Dz \, \sigma_l^2(\sqrt{q_l} z) \ , \tag{131}$$

$$= \frac{q^l}{2} \ , \tag{132}$$

$$\hat{q}_{xy}^l = \int Dz_1 \, Dz_2 \, \sigma_l(\sqrt{q_l} \, z_1 \gamma_l) \sigma_l \left[ \sqrt{q_l} \left( c_{xy}^l \, z_1 + \sqrt{1 - (c_{xy}^l)^2} \, z_2 \right) \gamma_l \right] \ , \tag{133}$$

$$= \frac{q^l}{2\pi} \left[ \sqrt{1 - c_{xy}^2} + \frac{c_{xy}\pi}{2} + c_{xy} \sin^{-1}(c_{xy}) \right] \ , \tag{134}$$

$$\hat{q}_{\alpha\beta,xy}^l = \int Dz_1 \, Dz_2 \, \sigma_l(\sqrt{q_l} \, z_1 \gamma_l) \sigma_l \left[ \sqrt{q_l} \left( c_{\alpha\beta,xy}^l \, z_1 + \sqrt{1 - (c_{\alpha\beta,xy}^l)^2} \, z_2 \right) \gamma_l \right] \ , \tag{135}$$

$$= \frac{q^l}{2\pi} \left[ \sqrt{1 - c_{\alpha\beta,xy}^2} + \frac{c_{\alpha\beta,xy}\pi}{2} + c_{\alpha\beta,xy} \sin^{-1}(c_{\alpha\beta,xy}) \right] \ . \tag{136}$$

$\square$

**Claim 6.6.** *The backward recursions are $\tilde{q}^l = \frac{\sigma_{\mathrm{w}}^2 \tilde{q}^{l+1}}{2}$, $\tilde{q}_{xy}^l = \frac{\sigma_{\mathrm{w}}^2 \tilde{q}_{xy}^{l+1}}{2\pi} \left[ \frac{\pi}{2} + \sin^{-1}(c_{xy}) \right]$ and $\tilde{q}_{\alpha\beta,xy}^l = \frac{\sigma_{\mathrm{w}}^2 \tilde{q}_{\alpha\beta,xy}^{l+1}}{2\pi} \left[ \frac{\pi}{2} + \sin^{-1}(c_{\alpha\beta,xy}) \right]$.*

*Derivation.* In the backward direction, we have

$$\delta_\alpha^l[i] = \sum_{(\beta,k) \in \mathcal{K}_{l+1} \times [C_{l+1}]} \frac{\partial z_\beta^{l+1}[k]}{\partial z_\alpha^l[i]} \delta_\beta^{l+1}[k] \ , \tag{137}$$

$$= \sigma_l'\left( z_\alpha^l[i] \right) \sum_{(\beta,k) \in \mathcal{F}_{l+1} \times [C_{l+1}]} [W_\beta^{l+1}]_{ki} \delta_{\alpha-\beta}^{l+1}[k] \tag{138}$$

Under the usual independence assumptions,

$$\mathbb{E}_\theta \langle \delta_\alpha^l(x), \delta_\beta^l(y) \rangle = \frac{\sigma_{\mathrm{w}}^2}{N_l} \mathbb{E}_\theta \left\langle \sigma_l'\left( z_\alpha^l(x) \right), \sigma_l'\left( z_\beta^l(y) \right) \right\rangle \sum_{\delta \in \mathcal{F}_{l+1}} \langle \delta_{\alpha-\delta}^{l+1}(x), \delta_{\beta-\delta}^{l+1}(y) \rangle \ . \tag{139}$$

Thus,

$$\tilde{q}^l := \mathop{\mathbb{E}}_{\alpha} \mathop{\mathbb{E}}_{x,\theta} \|\delta_\alpha^l(x)\|^2 \ , \tag{140}$$

$$= \frac{\sigma_{\mathrm{w}}^2 \tilde{q}^{l+1}}{2} \tag{141}$$

$$\tilde{q}_{xy}^l := \mathop{\mathbb{E}}_{\alpha} \mathop{\mathbb{E}}_{x,y,\theta} \langle \delta_\alpha^l(x), \delta_\alpha^l(y) \rangle \ , \tag{142}$$

$$= \frac{\sigma_{\mathrm{w}}^2 \tilde{q}_{xy}^{l+1}}{2\pi} \left( \frac{\pi}{2} + \sin^{-1} c_{xy} \right) \tag{143}$$

$$\tilde{q}_{\alpha\beta,xy}^l := \mathop{\mathbb{E}}_{\alpha \neq \beta} \left[ \mathop{\mathbb{E}}_{x,y,\theta} \langle \delta_\alpha^l(x), \delta_\beta^l(y) \rangle \right] \ , \tag{144}$$

$$= \frac{\sigma_{\mathrm{w}}^2 \tilde{q}_{\alpha\beta,xy}^{l+1}}{2\pi} \left( \frac{\pi}{2} + \sin^{-1} c_{\alpha\beta,xy} \right) \ . \tag{145}$$

$\square$

## 6.3 DERIVATION OF CLAIM 3.1

Recall that the Fisher information matrix $I_\theta \in \mathbb{R}^{n \times n}$ is given by

$$I_\theta = \mathop{\mathbb{E}}_{x \sim \mathcal{D}} \left[ \nabla_\theta f_\theta(x) \otimes \nabla_\theta f_\theta(x) \right] \ . \tag{146}$$

The claim is that the maximum eigenvalue of the Fisher Information Matrix is bounded as follows

$$\mathbb{E}_{(x,y) \sim \mathcal{D}} \langle \nabla_\theta f_\theta(x), \nabla_\theta f_\theta(y) \rangle \leq \lambda_{\max}(I_\theta) \leq \mathbb{E}_{x \sim \mathcal{D}} \langle \nabla_\theta f_\theta(x), \nabla_\theta f_\theta(x) \rangle \ . \tag{147}$$

The upper bound follows from convexity of $\lambda_{\max}(\cdot)$. Karakida et al. (2018) obtain the lower bound by considering the empirical estimate of the Fisher Information Matrix,

$$\widehat{I}_\theta = \frac{1}{m} \sum_{i=1}^m \nabla_\theta f_\theta(x_i) \otimes \nabla_\theta f_\theta(x_i) \ , \tag{148}$$

$$= \frac{1}{m} B_m^\top B_m \ . \tag{149}$$

where we have defined the matrix $B_m \in \mathbb{R}^{m \times n}$ with components $[B_m]_{ij} := \frac{\partial f_\theta(x_i)}{\partial \theta_j}$. We can then define a symmetric matrix $\widehat{F}_\theta \in \mathbb{R}^{m \times m}$ with the same eigenvalues as $\widehat{I}_\theta$,

$$\widehat{F}_\theta = \frac{1}{m} B_m B_m^\top \ , \tag{150}$$

The maximal eigenvalue of $\widehat{I}_\theta$ is thus computed by the Rayleigh quotient,

$$\lambda_{\max}(\widehat{I}_\theta) = \lambda_{\max}(\widehat{F}_\theta) = \max_{v : \|v\|=1} \langle v, \widehat{F}_\theta v \rangle \ . \tag{151}$$

Letting $\mathbf{1} \in \mathbb{R}^m$ denote the vector of ones,

$$\lambda_{\max}(\widehat{I}_\theta) \geq \left\langle \frac{1}{\sqrt{m}} \mathbf{1}, \widehat{F}_\theta \frac{1}{\sqrt{m}} \mathbf{1} \right\rangle \ , \tag{152}$$

$$= \frac{1}{m^2} \sum_{i,j \in [m]} \langle \nabla_\theta f_\theta(x_i), \nabla_\theta f_\theta(x_j) \rangle, \tag{153}$$

which is the the plug-in estimator (V-statistic). Thus, for a layered neural network we have

$$\bar{\lambda}_{\max}(I_\theta) \geq \frac{1}{m^2} \sum_{(x,y)} \sum_{l \in [L]} \langle \nabla_{\theta_l} f_\theta(x), \nabla_{\theta_l} f_\theta(y) \rangle \ , \tag{154}$$

$$=: \sum_{l \in [L]} f_l. \tag{155}$$

Specializing to a fully-connected neural network we obtain,

$$f_l = \frac{1}{m^2} \sum_{(x,y)} \left[ \left\langle \frac{\partial f_\theta(x)}{\partial b_l}, \frac{\partial f_\theta(y)}{\partial b_l} \right\rangle + \left\langle \frac{\partial f_\theta(x)}{\partial W_l}, \frac{\partial f_\theta(y)}{\partial W_l} \right\rangle \right. \tag{156}$$

$$\left. + \left\langle \frac{\partial f_\theta(x)}{\partial \gamma_l}, \frac{\partial f_\theta(y)}{\partial \gamma_l} \right\rangle + \left\langle \frac{\partial f_\theta(x)}{\partial \beta_l}, \frac{\partial f_\theta(y)}{\partial \beta_l} \right\rangle \right] \;, \tag{157}$$

$$\simeq \frac{1}{m^2} \sum_{(x,y)} \left\langle \frac{\partial f_\theta(x)}{\partial W_l}, \frac{\partial f_\theta(y)}{\partial W_l} \right\rangle \;, \tag{158}$$

$$= \frac{1}{m^2} \sum_{(x,y)} \langle h^{l-1}(x), h^{l-1}(y) \rangle \langle \delta^l(x), \delta^l(y) \rangle \;, \tag{159}$$

$$\simeq \left[ \frac{1}{m^2} \sum_{(x,y)} \langle h^{l-1}(x), h^{l-1}(y) \rangle \right] \left[ \frac{1}{m^2} \sum_{(x,y)} \langle \delta^l(x), \delta^l(y) \rangle \right] \;, \tag{160}$$

$$= N_{l-1} \hat{q}_{xy}^{l-1} \tilde{q}_{xy}^l \;. \tag{161}$$

In the first approximation we use the fact that the terms with respect to $b, \gamma, \beta$ are $N_l$ times smaller than the term with respect to $W$. The second approximation uses the assumption that forward and backward order parameters are independent. The last approximation is for $m \gg 1$.

For convolutional layers we have

$$f_l = \frac{1}{m^2} \sum_{(x,y)} \left[ \left\langle \frac{\partial f_\theta(x)}{\partial b_l}, \frac{\partial f_\theta(y)}{\partial b_l} \right\rangle + \left\langle \frac{\partial f_\theta(x)}{\partial W_l}, \frac{\partial f_\theta(y)}{\partial W_l} \right\rangle \right. \tag{162}$$

$$\left. + \left\langle \frac{\partial f_\theta(x)}{\partial \gamma_l}, \frac{\partial f_\theta(y)}{\partial \gamma_l} \right\rangle + \left\langle \frac{\partial f_\theta(x)}{\partial \beta_l}, \frac{\partial f_\theta(y)}{\partial \beta_l} \right\rangle \right] \;, \tag{163}$$

$$\simeq \frac{1}{m^2} \sum_{(x,y)} \left\langle \frac{\partial f_\theta(x)}{\partial W_l}, \frac{\partial f_\theta(y)}{\partial W_l} \right\rangle \;, \tag{164}$$

$$= \frac{1}{m^2} \sum_{(x,y)} \sum_{\alpha \in \mathcal{F}_l} \sum_{\beta,\beta' \in \mathcal{K}_l} \langle \delta_\beta^l(x), \delta_{\beta'}^l(y) \rangle \langle h_{\alpha+\beta}^{l-1}(x), h_{\alpha+\beta'}^{l-1}(y) \rangle \;, \tag{165}$$

$$\simeq \frac{1}{|\mathcal{K}_l|^2} \left[ \frac{1}{m^2} \sum_{(x,y)} \sum_{\beta,\beta' \in \mathcal{K}_l} \langle \delta_\beta^l(x), \delta_{\beta'}^l(y) \rangle \right] \left[ \frac{1}{m^2} \sum_{(x,y)} \sum_{\alpha \in \mathcal{F}_l} \sum_{\beta,\beta' \in \mathcal{K}_l} \langle h_{\alpha+\beta}^{l-1}(x), h_{\alpha+\beta'}^{l-1}(y) \rangle \right] \;, \tag{166}$$

$$= \left[ |\mathcal{K}_l|(|\mathcal{K}_l| - 1)\tilde{q}_{\alpha\beta,xy}^l + |\mathcal{K}_l|\tilde{q}_{xy}^l \right] \left[ \frac{\mathcal{C}_{l-1}|\mathcal{F}_l||\mathcal{K}_l|(|\mathcal{K}_l| - 1)}{|\mathcal{K}_l|^2} \hat{q}_{\alpha\beta,xy}^{l-1} + \frac{\mathcal{C}_{l-1}|\mathcal{K}_l||\mathcal{F}_l|}{|\mathcal{K}_l|^2} \hat{q}_{xy}^{l-1} \right] \;, \tag{167}$$

$$= N_{l-1} \left[ (|\mathcal{K}_l| - 1)\tilde{q}_{\alpha\beta,xy}^l + \tilde{q}_{xy}^l \right] \left[ (|\mathcal{K}_l| - 1)\hat{q}_{\alpha\beta,xy}^{l-1} + \hat{q}_{xy}^{l-1} \right] \;, \tag{168}$$

where $N_{l-1} = \mathcal{C}_{l-1}|\mathcal{F}_l|$. In the first approximation we use the fact that the terms with respect to $b, \gamma, \beta$ are $c_l$ times smaller than the term with respect to $W$. The second approximation uses the assumption that forward and backward order parameters are independent.

## 6.4 BASELINE

Baseline was an experiment of vanilla fully-connected networks trained on MNIST, with various $\sigma_w$ weight initializations. Result is shown in Fig. 3.

## 6.5 ADDITIONAL EXPERIMENTS

Our theory predicts that smaller $\gamma$ has the effect of greatly reducing the $\lambda_{max}$ of the FIM, and empirically networks with BatchNorm converge faster. Following this intuition, we performed additional

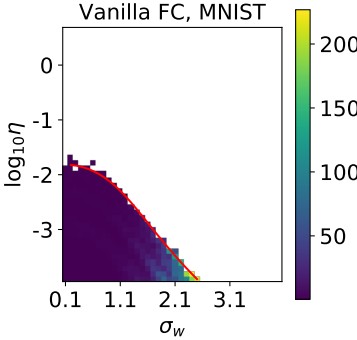

Figure 3: Heatmap showing test loss as a function of $(\log_{10}\eta, \gamma)$ after 5 epochs of training for vanilla fully-connected feed forward network where the relation between weight initialization variance and maximal learning rate is studied.

experiments with VGG16 Simonyan & Zisserman (2014) and Preact-Resnet18 He et al. (2016a), with various $\gamma$ initializations, fixed learning rate 0.1, momentum 0.9 and weight decay 0.0005, trained on CIFAR-10. The result is average over 5 different independent trainings. We find that smaller $\gamma$ initialization indeed increases the speed of convergence, as shown below in Fig. 4.

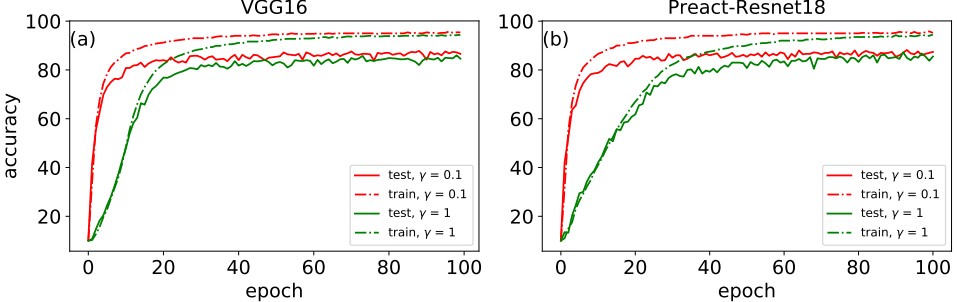

Figure 4: Learning curves of (a) VGG16 and (b) Preact-Resnet18 training on CIFAR-10, with the same hyperparameters except $\gamma$ initialization. Results support that small $\gamma$ initialization helps faster convergence.

