# OpenReview forum: "MEAN-FIELD ANALYSIS OF BATCH NORMALIZATION"
_ICLR.cc/2019/Conference_

### Official Review · AnonReviewer3 · 2018-10-28
**Theoretical but not rigorous**

**Rating:** 5
**Confidence:** 3

**Review:**

In this paper, the effect of batch normalization to the maximum eigenvalue of the Fisher information is analyzed. The techinique is mostly developed by Karakida et al. (2018). The main result is an informal bound of the maximum eigenvalue, which is given without proof. Though, the numerical result corresponds to the derived bound.

The paper is basically well written, but the technical part has several notational problems. For example, there is no definition of "\otimes", "\odot", and "Hess" operators.

The use of the mean-field theory is an interesting direction to analyze batch normalization. However, in this paper, it seems failed to say some rigorous conclusion. Indeed, all of the theoretical outcomes are written as "Claims" and no formal proof is given. Also, there is no clear explanation of why the authors give the results in a non-rigorous way, where is the difficult part to analyze in a rigorous way, etc.

Aside from the rigor issue, the paper heavily depends on the study of Karakida et al. (2018). The derivation of the bound (44) is directly built on Karakida's results such as Eqs. (7,8,20--22), which reduces the paper's originality.

The paper also lacks practical value. Can we improve an algorithm or something by using the bound (44) or other results?

---

> ### Author Response · Authors · 2018-11-11
> **Thanks for your review! Additional experiments and results have been added. Part 2**
>
> 4. Aside from the rigor issue, the paper heavily depends on the study of Karakida et al. (2018). The derivation of the bound (44) is directly built on Karakida's results such as Eqs. (7,8,20--22), which reduces the paper's originality.
>     The paper also lacks practical value.  Can we improve an algorithm or some-thing by using the bound (44) or other results?
>
>    Although our paper is motivated by their approach, Karakida et al. (2018) have different goals than us, and we significantly extend the framework to address our questions. While Karakida et al. (2018) focuses on studying the statistics of the FIM for vanilla (no BatchNorm) fully-connected neural networks, our aim is to study the role of BatchNorm. Therefore we extend the theory significantly, to both fully-connected and convolutional neural networks, with and without BatchNorm, and derive a new lower bound for ConvNets. We find that adding BatchNorm can greatly reduce the maximal eigenvalue of the FIM, and perform experiments to verify this.
>
>     A practical upshot of the paper is that faster convergence is linked to smaller \gamma-initialization, which is a new practical finding to our knowledge. To justify this, we have performed additional experiments in the updated version of our paper with VGG16 and Preact-Resnet18 with various \gamma initializations trained on CIFAR-10. We find that a smaller \gamma initialization indeed increases the speed of convergence. This result is included in the SM of the latest version of our paper. Thus, we believe that our work has both theoretical and practical value that should be of use to other researchers.
>
>     More generally, by excluding unfeasible regions of parameters space, our analysis can be used for  hyperparameter search in more realistic architectures than the fully-connected ones considered in Karakida.
>
> Thank you again for your review and comments. Hopefully our reply has addressed your question and concerns.
>
> [1]Samuel S Schoenholz, Justin Gilmer, Surya Ganguli, and Jascha Sohl-Dickstein. Deep informationpropagation. In International Conference on Learning Representations (ICLR), 2017.
> [2]Lechao Xiao, Yasaman Bahri, Jascha Sohl-Dickstein, Samuel S. Schoenholz, and Jeffrey Penning-ton. Dynamical isometry and a mean field theory of cnns:  How to train 10,000-layer vanilla convolutional neural networks. In International Conference on Machine Learning (ICML), 2018.

---

> ### Author Response · Authors · 2018-11-11
> **Thanks for your review! Additional experiments and results have been added. Part 1**
>
> Thank you very much for your review and helpful comments. We address your specific questions and comments below:
>
> 1. The main result is an informal bound of the maximum eigenvalue, which is given without proof. Though, the numerical result corresponds to the derived bound.
>
> We omitted some important steps in the proof of the bound for the maximum eigenvalue in the original version. We have updated the detailed proof in the SM of our latest version, and apologize for any confusion this caused.
>
> 2. The paper is basically well written, but the technical part has several notational problems. For example, there is no definition of "$\otimes$", "$\odot$", and "Hess" operators.
>
> Thanks for the comments. We have updated the paper and added definitions and explanations for all notations.
>
> 3. The use of the mean-field theory is an interesting direction to analyze batch normalization. However, in this paper, it seems failed to say some rigorous conclusion. Indeed, all of the theoretical outcomes are written as "Claims" and no formal proof is given. Also, there is no clear explanation of why the authors give the results in a non-rigorous way, where is the difficult part to analyze in a rigorous way, etc.
>
>  Thanks for raising this issue, and allow us an attempt to clarify. Our approach to estimating the maximal eigenvalue of the FIM for a random neural network involves two assumptions. First, we assume a large layer width in the network so that the behavior of a hidden node can be approximated by Gaussian distribution due to central limit theorem. Second, we assume that the averages for the forward and backward pass in the network are uncorrelated. Both assumptions are common and empirically successful in existing literature on mean-field theory of neural networks[1][2], however the second one in particular lacks a rigorous justification. Therefore we present our results as claims instead of theorems to emphasize that additional work is needed to rigorously justify the existing assumptions in the mean field literature generally.
>
>     To make as explicit as possible our assumptions mentioned above, we have added a clear derivation in our latest version that hopefully will give the reader greater confidence in the rigor of our results.
>
>     In addition, we acknowledge that the assumptions stated above have not been rigorously justified, albeit being well-accepted in other papers. Thus we performed extensive experiments to test the validity of our theoretical results, finding that indeed the experiments correspond strikingly well to the theory.

---

### Official Review · AnonReviewer1 · 2018-11-02
**Interesting application of MFT on FIM to understand Batch Normalization**

**Rating:** 6
**Confidence:** 3

**Review:**

Interesting application of MFT on FIM to understand Batch Normalization

This paper applies mean field analysis to networks with batch normalization layers. Analyzing maximum eigenvalue of the Fisher Information Matrix, the authors provide theoretical evidence of allowing higher learning rates and faster convergence of networks with batch normalization.

The analysis reduces to providing lower bound for maximum eigenvalue of FIM using mean-field approximation. Authors provide lower bound of the maximum eigenvalue in the case of fully-connected and convolutional networks with batch normalization layers. Lastly authors observe empirical correlation between smaller \gamma and lower test loss.

Pro:
 - Clear result providing theoretical ground for commonly observed effects.
 - Experiments are simple but illustrative. It is quite surprising how well the maximum learning rate prediction matches with actual training performance curve.


Con:
 - While mean field analysis a-priori works in the limit where networks width goes to infinity for fixed dataset size, the analysis of Fisher and Batch normalization need asymptotic limit of dataset size.
 - Although some interesting results are provided. The content could be expanded further for conference submission. The prediction on maximum learning rate is interesting and the concrete result from mean field analysis
 - While correlation between batch norm \gamma parameter and test loss is also interesting, the provided theory does not seem to provide good intuition about the phenomenon.

Comments:
- The theory provides the means to compute lower bound of maximum eigenvalue of FIM using mean-field theory. In Figure 1, is \bar \lambda_{max} computed using the theory or empirically computed on the actual network? It would be nice to make this clear.
- In Figure 2, the observed \eta_*/2 of dark bands in heatmap is interesting. While most of networks without Batch Norm, performance is maximized using learning rates very close to maximal value, often networks using batch norm the learning rate with maximal performance is not the maximal one and it would be interesting to provide theoretical
- I feel like section 3.2 should cite Xiao et al (2018). Although this paper is cited in the intro, the mean field analysis of convolutional layers was first worked out in this paper and should be credited.

---

> ### Author Response · Authors · 2018-11-11
> **Thanks for your review! Additional experiments and results have been added.**
>
> Thank you very much for your review and helpful comments. We address your questions and concerns individually below:
>
> 1. While mean field analysis a priori works in the limit where networks width goes to infinity for fixed dataset size, the analysis of Fisher and Batch normalization need asymptotic limit of dataset size.
>
> Thank you for pointing this out. Our derivation of Claim 3.1 from (153) to (154) in SM is based on older definitions of order parameters, where E_{x, y} was replaced by E_{x \neq y}, and therefore the asymptotic limit of large dataset size was required.
>
> However, based on our new definitions of order parameters, (153) to (155) are exact, and we should have removed (154) and revised Claim 3.1. Therefore in our new version, the asymptotic limit of large dataset size is not required in Claim 3.1. We apologize for this mistake and concomitant confusion.
>
> Derivation of recursion relation also requires large dataset size, m, where the error for finite m is O(1/m). Therefore even for a dataset of size 100, the error is around 1%, and the error introduced by finite m is negligible for most of the frequently-used datasets. We have added an explanation of this issue in the latest version of the submission.
>
> The other place where there is a potential issue of large dataset size is in using the empirical FIM to approximate the true FIM in Section 2.1. However, since we are concerned here with the convergence of the learning dynamics on the training set, the empirical FIM is actually sufficient for our analysis. For future work on extending this theory to study generalization, limited dataset size must be taken into account.
>
> 2. Although some interesting results are provided. The content could be expanded further for conference submission. The prediction on maximum learning rate is interesting and the concrete result from mean field analysis...[did this get cut off?]
> While correlation between batch norm \gamma parameter and test loss is also interesting, the provided theory does not seem to provide good intuition about the phenomenon.
>
> Indeed, this is correct. Our approach targets exploring the change of the FIM spectrum, and hence the maximal learning rate, with/without BatchNorm, and hence isn't able to directly make statements about generalization. However, our theory predicts that faster convergence is linked to smaller \gamma-initialization, which is a new practical finding to our knowledge. Following this intuition, we performed additional experiments in the updated version of our paper with VGG16 and Preact-Resnet18, with various \gamma initializations, trained on CIFAR-10. We find that the smaller \gamma initialization indeed increase the speed of convergence. This result can be found in the SM of the latest version of our paper.
>
> 3. The theory provides the means to compute lower bound of maximum eigenvalue of FIM using mean-field theory. In Figure 1, is \lambda_{max} computed using the theory or empirically computed on the actual network? It would be nice to make this clear.
>
> We are sorry for this confusion. It is computed using the theory and we have clarified this in our latest version. This is also useful in practice because direct numerical calculation of \lambda_max is difficult for realistic deep neural networks due to high computational cost.
>
> 4. In Figure 2, the observed \eta_*/2 of dark bands in heatmap is interesting. While most of networks without Batch Norm, performance is maximized using learning rates very close to maximal value, often networks using batch norm the learning rate with maximal performance is not the maximal one and it would be interesting to provide theoretical.
>
> This is indeed an interesting observation, but since our theory can't directly speak to performance (it analyzes the maximal allowed rate instead of the optimal rate), a different approach would be required to explain this phenomenon.
>
> 5. I feel like section 3.2 should cite Xiao et al (2018). Although this paper is cited in the intro, the mean field analysis of convolutional layers was first worked out in this paper and should be credited.
>
> Yes certainly, and we apologize for the oversight. We have updated the citation in our latest version.
>
> Thank you again for your review and comments. Hopefully our reply has addressed your question and concerns.

---

> > ### Comment · AnonReviewer1 · 2018-12-05
> > **thanks for the clarifications**
> >
> > I thank the authors for providing answers to raised questions and clarifications. Also I appreciate the efforts to make the revisions.
> >
> > -- "Derivation of recursion relation also requires large dataset size, m, where the error for finite m is O(1/m). Therefore even for a dataset of size 100, the error is around 1%, and the error introduced by finite m is negligible for most of the frequently-used datasets."
> >
> > I might still worry about constant factor multiplying 1/m and would happy to see this effect is indeed suppressed sufficiently.
> >
> > extra typo: Figure 3 caption should be (\log_{10} \eta, \sigma_w)
> >
> > Also original VGG-16 does not have batch-norm, and it should be made clear that the experiments were done on the modified version of VGG-16.

---

> > > ### Author Response · Authors · 2018-12-06
> > > **thanks for the response**
> > >
> > > We thank Reviewer1 for the response. We have performed additional experiments and further address your questions below:
> > >
> > > 1. I might still worry about constant factor multiplying 1/m and would happy to see this effect is indeed suppressed sufficiently.
> > >
> > > In other to see the error suppressed by dataset size m, we performed additional experiments on finding maximal learning rate of fully-connected NN with MNIST and ConvNet with CIFAR10, where training dataset size m varies from 5 to 50000 and dataset is randomly sampled from the original dataset. The results are shown as below:
> > >
> > > fully-connected on MNIST, \gamma = 0.5
> > > --------------------------------------------------------------------------------------------------------
> > >        m        |    5    |    10   |   50   |   100  |  500  | 1000 | 5000 |10000| 50000
> > > ---------------------------------------------------------------------------------------------------------
> > > log10(eta)| -1.39 | -1.37 | -1.32 | -1.32 | -1.31 | -1.32 | -1.32 | -1.32 | -1.32
> > > --------------------------------------------------------------------------------------------------------
> > >
> > > fully-connected on MNIST, \gamma = 1
> > > --------------------------------------------------------------------------------------------------------
> > >        m        |    5    |    10   |   50   |   100  |  500  | 1000 | 5000 |10000| 50000
> > > ---------------------------------------------------------------------------------------------------------
> > > log10(eta)| -2.20 | -1.99 | -1.92 | -1.91 | -1.91 | -1.91 | -1.91 | -1.91 | -1.91
> > > --------------------------------------------------------------------------------------------------------
> > >
> > > ConNet on CIFAR10, \gamma = 0.5
> > > --------------------------------------------------------------------------------------------------------
> > >        m        |    5    |    10   |   50   |   100  |  500  | 1000 | 5000 |10000| 50000
> > > ---------------------------------------------------------------------------------------------------------
> > > log10(eta)| -1.30 | -1.26 | -1.25 | -1.24 | -1.24 | -1.24 | -1.24 | -1.24 | -1.24
> > > --------------------------------------------------------------------------------------------------------
> > >
> > > ConvNet on CIFAR10, \gamma = 1
> > > --------------------------------------------------------------------------------------------------------
> > >        m        |    5    |    10   |   50   |   100  |  500  | 1000 | 5000 |10000| 50000
> > > ---------------------------------------------------------------------------------------------------------
> > > log10(eta)| -1.91 | -1.85 | -1.82 | -1.83 | -1.82 | -1.83 | -1.82 | -1.82 | -1.82
> > > --------------------------------------------------------------------------------------------------------
> > >
> > > notice that we used step size of 0.01 when scanning the learning rate values to find the maximal learning rate. We observe that maximal learning rate is increasing with dataset size m when m < 50 and becomes stable and saturated when m > 50 for all cases. These experiments are strong evidence that the error introduced by limited data is indeed suppressed sufficiently in most of the dataset we are interested in.
> > >
> > > We hope the addition experiments can address your concern and we will include them in the final version.
> > >
> > > 2. extra typo: Figure 3 caption should be (\log_{10} \eta, \sigma_w) . Also original VGG-16 does not have batch-norm, and it should be made clear that the experiments were done on the modified version of VGG-16.
> > >
> > > We apologize for the confusion and we will update it in the final version.
> > >
> > > Thank you again for your review and comments, we hope our response address your concerns.

---

### Official Review · AnonReviewer2 · 2018-11-02
**Interesting paper**

**Rating:** 7
**Confidence:** 3

**Review:**

This paper studies the effect of batch normalization via a physics style mean-field theory. The theory yields a prediction of maximal learning rate for fully-connected and convolutional networks, and experimentally the max learning rate agrees very well with the theoretical prediction.

This is a well-written paper with a clean, novel result: when we fix the BatchNorm parameter \gamma, a smaller \gamma stabilizes the training better (allowing a greater range of learning rates). Though in practice the BatchNorm parameters are also trained, this result may suggest using a smaller initialization.

A couple of things I was wondering:

-- As a baseline, how would the max learning rate behave without BatchNorm? Would the theories again match the experimental result there?

-- Is the presence of momentum important? If I set the momentum to be zero, it does not change the theory about the Fisher information and only affects the dependence of \eta on the Fisher information. In this case would the theory still match the experiments?

---

> ### Author Response · Authors · 2018-11-11
> **Thanks for your review! Additional experiments and results have been added.**
>
> Thank you very much for your review and valuable comments. We address your questions and comments below:
>
> 1. As a baseline, how would the max learning rate behave without BatchNorm? Would the theories again match the experimental result there?
>
> We also wondered how the max learning rate would behave without BatchNorm, and thus we did an experiment for a network without BatchNorm where we varied \sigma_w, the weight initialization variance, and found that the theory again matches the experimental result. However, we didn’t include this result in the previous draft. We have now added this result to the SM in the new revised version as a baseline.
>
> 2. Is the presence of momentum important? If I set the momentum to be zero, it does not change the theory about the Fisher information and only affects the dependence of $\eta$ on the Fisher information. In this case would the theory still match the experiments?
>
> The presence of momentum doesn't change the picture dramatically. We set momentum to 0.9 to match the value frequently used in practice. Indeed, changing the momentum only affects the dependency of \eta on the FIM. We have performed an additional experiment on training without momentum and find that in this case the theory still matches the experiment.
>
> 3. This is a well-written paper with a clean, novel result: when we fix the BatchNorm parameter \gamma, a smaller \gamma stabilizes the training better (allowing a greater range of learning rates). Though in practice the BatchNorm parameters are also trained, this result may suggest using a smaller initialization.
>
> Thanks for the positive feedback! We performed additional experiments in the updated version of our paper with VGG11 and Preact-Resnet18, with various \gamma-initializations, trained on CIFAR-10.  We find that the smaller \gamma-initialization indeed increase the speed of convergence.  This result can be found in the SM of the latest version of our paper.
>
> Thank you again for your review and comments. We believe that the inclusion of a baseline without BatchNorm as well as clarification on the role of momentum has improved the results and clarity of the paper.

---

### Meta-Review · Area_Chair1 · 2018-12-10
**a promising start, but the analysis is mechanical and the maximum learning rate isn't inherently meaningful**

**Confidence:** 5
**Recommendation:** Reject

**Metareview:**

This paper presents a mean field analysis of the effect of batch norm on optimization. Assuming the weights and biases are independent Gaussians (an assumption that's led to other interesting analysis), they propagate various statistics through the network, which lets them derive the maximum eigenvalue of the Fisher information matrix. This determines the maximum learning rate at which learning is stable. The finding is that batch norm allows larger learning rates.

In terms of novelty, the paper builds on the analysis of Karakida et al. (2018). The derivations are mostly mechanical, though there's probably still sufficient novelty.

Unfortunately, it's not clear what we learn at the end of the day. The maximum learning rate isn't very meaningful to analyze, since the learning rate is only meaningful relative to the scale of the weights and gradients, and the distance that needs to be moved to reach the optimum. The authors claim that a "higher learning rate leads to faster convergence", but this seems false, and at the very least would need more justification. It's well-known that batch norm rescales the norm of the gradients inversely to the norm of the weights; hence, if the weight norm is larger than 1, BN will reduce the gradient norm and hence increase the maximum learning rate. But this isn't a very interesting effect from an optimization perspective. I can't tell from the analysis whether there's a more meaningful sense in which BN speeds up convergence. The condition number might be more relevant from a convergence perspective.

Overall, this paper is a promising start, but needs more work before it's ready for publication at ICLR.